# Do Patients with Bronchiectasis Have an Increased Risk of Developing Lung Cancer? A Systematic Review

**DOI:** 10.3390/life13020459

**Published:** 2023-02-07

**Authors:** Nadia Castaldo, Alberto Fantin, Massimiliano Manera, Vincenzo Patruno, Giulia Sartori, Ernesto Crisafulli

**Affiliations:** 1Department of Pulmonology, University Hospital of Udine (ASUFC), 33100 Udine, Italy; 2Respiratory Medicine Unit, Department of Medicine, University of Verona and Azienda Ospedaliera Universitaria Integrata of Verona, 37126 Verona, Italy

**Keywords:** bronchiectasis, lung cancer, COPD, hazard risk

## Abstract

**Simple Summary:**

A growing amount of evidence points out that patients with chronic respiratory disease have an increased risk of lung cancer. We conducted a systematic review of all published data to define the characteristics of lung malignancies in patients with non-cystic fibrosis bronchiectasis and the characteristics of patients who develop bronchiectasis-associated lung cancer. The frequency rates of lung cancer in patients with non-cystic fibrosis bronchiectasis (NCFB) ranged from 0.93% to 8.0%. The incidence rate of lung cancer in bronchiectasis patients was 3.96, and adenocarcinoma was the most frequent histological type. The primary cause of death in the bronchiectasis group was malignancy (31.2%), particularly lung cancer (12.4%). We found that the presence of NCFB was associated with a higher risk of developing lung cancer than the population without NCFB, and this risk was higher for males, the elderly, and smokers. However, the effect of the co-existence of bronchiectasis and chronic obstructive pulmonary disease was unclear.

**Abstract:**

Background: Initial evidence supports the hypothesis that patients with non-cystic fibrosis bronchiectasis (NCFB) have a higher risk of lung cancer. We systematically reviewed the available literature to define the characteristics of lung malignancies in patients with bronchiectasis and the characteristics of patients who develop bronchiectasis-associated lung cancer. Method: This study was performed based on the PRISMA guidelines. The review protocol was registered in PROSPERO. Results: The frequency rates of lung cancer in patients with NCFB ranged from 0.93% to 8.0%. The incidence rate was 3.96. Cancer more frequently occurred in the elderly and males. Three studies found an overall higher risk of developing lung cancer in the NCFB population compared to the non-bronchiectasis one, and adenocarcinoma was the most frequently reported histological type. The effect of the co-existence of NCFB and COPD was unclear. Conclusions: NCFB is associated with a higher risk of developing lung cancer than individuals without NCFB. This risk is higher for males, the elderly, and smokers, whereas concomitant COPD’s effect is unclear.

## 1. Introduction

Adult non-cystic fibrosis bronchiectasis (NCFB) is the third most common chronic inflammatory respiratory disease, after chronic obstructive pulmonary disease (COPD) and asthma [1]. However, in the past, the clinical burden of NCFB had been profoundly underestimated. Notwithstanding, NCFB is a complex disease characterized by chronic systemic inflammation that frequently co-exists with comorbidities and several respiratory conditions, including pulmonary hypertension, COPD, severe asthma, recurrent respiratory infections, and lung, esophageal, and hematological malignancies [2,3,4].

The pathophysiology of NCFB consists of a vicious cycle of persistent inflammation, recurrent airway infections, and structural and functional damage of the airways, causing the progression of the disease. The main feature of NCFB is the heterogeneity in clinical presentation (phenotypes) and pathophysiological mechanisms of occurrence (endotypes). The existence of different endotypes of NCFB is related to different molecular pathways [5]. In fact, the recent advances of molecular biology technologies have allowed the determination of several markers related to NCFB endotypes (e.g., NCFB related to neutrophil dysfunction and protease-anti protease unbalance [6], neutrophil extracellular traps [7], Eosinophilic/Type 2 Inflammatory Endotype [8]). The identification of some of these subsets of patients may lead to tailored treatment (“treatable trait”) [9].

The clinical overlap between the multiple respiratory affections, the heterogeneous clinical presentation, and the frequently inadequate radiological interpretation lead to the misdiagnosis and underdiagnoses of NCFB [10]. Although the exact epidemiology of NCFB is still unknown, different authors have reported increasing prevalence and incidence rates of NCFB during the last three decades. For example, a large epidemiological analysis conducted in the United Kingdom (U.K.), found the prevalence of bronchiectasis increased from 2004 to 2013 in both women and men (from 350.5 per 100,000 to 566.1 per 100,000 person-years among women, and from 301.2 per 100,000 to 485.5 per 100,000 in 2013 among men) [11]. Like the U.K., the overall prevalence of NCFB in the United States of America (U.S.) increased from 812 cases per 100,000 inhabitants in 2000 to 1100 cases per 100,000 inhabitants in 2013 [11]. Overall, in the U.S., annual growth in prevalence rates of NCBF has been estimated to be 8.74%, especially in the elderly [12] This trend is not surprising since the wide use of high-resolution computed tomography (HRCT) of the chest (e.g., in lung cancer screening programs) has improved NCFB diagnosis [13].

Mortality rates are substantially higher in people with NCFB compared to the general population in all age groups. Overall, mortality rates in NCFB patients range from 10 to 20% over a 5–10 year follow-up period [14,15]. The primary cause of death is respiratory failure; the main risk factors for mortality are *Pseudomonas aeruginosa* airway colonization, low body mass index, male sex, older age, and co-existence with COPD [14,15]. Infective complications and comorbidities accompanying NCFB are the main determinants of the outcomes and the health burden of NCFB. In addition, patients with NCFB require more hospitalizations, extended hospital stays, and more frequent outpatient visits than their non-bronchiectasis counterparts [16]. Therefore, NCFB also implies higher medical costs [17,18].

Furthermore, a growing amount of evidence points out that patients with chronic respiratory disease, including COPD, idiopathic pulmonary fibrosis (IPF), and NCFB, have an increased risk of lung cancer [19,20,21,22,23,24,25,26,27,28]. Moreover, some authors have found lung cancer is associated with higher mortality in patients with NCFB [28]. However, no comprehensive studies or systematic reviews of the relationship between NCFB and lung cancer have been published. Therefore, we conducted this systematic review to (1) summarize the existing literature regarding the association between lung cancer and NCFB; (2) describe the characteristics of existing studies. Findings from this study are expected to inform the experts about information gaps and unmet needs regarding this setting to inspire additional work for future research.

The representative histology images of NCFB, COPD, and adenocarcinoma are showed in Figure 1, Figure 2 and Figure 3.

## 2. Materials and Methods

### 2.1. Study Protocol and Search Strategy

The present study has been conducted following the Preferred Reporting Items for Systematic reviews and Meta-Analyses (PRISMA) statement [29]. Five steps have been followed: (i) the design and search strategy; (ii) the collection of articles and their systematic review; (iii) the assessment of the inclusion and exclusion criteria; (iv) the qualitative assessment; and (v) statistical analysis of the data. Two researchers have carried out all stages independently (AF and NC). The study protocol was registered in the PROSPERO prospective international register of systematic reviews (CRD42022362754).

The authors conducted comprehensive literature research through international databases, such as PubMed/Medline, Scopus, WebOfScience, OVID, and COCHRANE Library. All the articles regarding lung cancer and bronchiectasis published before 31 July 2022, were searched using common keywords and Mesh and Mesh Entry. Results were limited to the English language and humans. The complete search strategy can be found in Appendix A. In addition, a manual search was also carried out to select relevant articles included in the reference lists of previously identified manuscripts.

We included all retrospective and prospective studies investigating lung cancer occurrence in NCFB patients. In addition, the following exclusion criteria were used: (1) letters, editorials, expert opinions, case reports or case series with less than ten patients, and reviews; (2) non-human studies; (3) duplicated or overlapped data; and (4) studies including cystic fibrosis-related bronchiectasis, due to the clinical, pathophysiologic, and inflammatory peculiarities of this condition.

### 2.2. Selection of Studies

After the initial screening process, all the studies retrieved from the literature search were exported to Endnote (V. 20.0), and duplicates were removed using the Endnote deduplication tool. In addition, to aid the screening and selection process, the deduplicated studies have been uploaded into HubMeta, a web-based data entry system (Hubmeta, Beta, 2020, available online: hubmeta.com). Hence, two reviewers (AF and NC) independently screened each study to select those eligible for inclusion in the review.

After selecting relevant studies, the authors examined the quality of the final studies selected for inclusion. The Newcastle–Ottawa Scale (NOS) checklist was used. Therefore, the studies were divided into three levels: (1) studies of poor quality (score of 5); (2) studies of medium quality (score of 5 to 6); (3) studies of high quality (score of 7 to 8) [30] (data not shown). Disagreements about study selection were discussed, and a third researcher (VP) was involved in achieving a consensus.

### 2.3. Data Synthesis

The authors’ initial purpose was to conduct a systematic review and meta-analysis. However, the heterogeneous nature of the included studies and the data underlying their conclusions did not allow for this analysis. Therefore, for this manuscript, the authors used a framework analysis approach to summarize the evidence of the association between lung cancer and NCFB [31].

Data from the included studies were extracted into a data extraction form. The following information was extracted and recorded: author(s), year of publication, type of study, country, sample size (SS), periods, the prevalence of lung cancer in NCFB (when available), the incidence of lung cancer in NCFB (when available), cellular (histological) subtypes of lung cancers, clinical staging, age, the prevalence by gender, the prevalence by smoking status, the prevalence of respiratory and non-respiratory comorbidities, the prevalence of immunodeficiency, the prevalence of chronic infections, and the region and location of lung cancer and NCFB. In addition to the abovementioned variables, the authors computed some additive variables based on availability in text or tables.

Data analysis was performed using IBM SPSS Statistics v 20 [32], and the significance level of the test was <0.05.

## 3. Results

### 3.1. Characteristics of Included Studies

The database search identified 7331 records. These records were exported to Endnote (Version 20.0, Clarivate Analytics, Spring Garden, Philadelphia, PA, USA), and 2103 duplicates were removed using the Endnote deduplication feature. Additionally, nine records were identified through hand searching. Therefore, 5228 unique citations were found across all database searches. Next, the title and abstract of the 5228 articles were screened, electing ten articles for full-text evaluation. After evaluating articles in full text, eight were deemed eligible for the review (see Figure 1).

The included studies (*n* = 8) were published between 2014 and 2022 and were retrospective [19,20,21,23,24,25,27,28]. Three studies were population-based cohorts of which two focused on risk factors of lung cancer in patients with NCFB [19,23], and one analyzed the prevalence and burden of NCFB in a lung cancer screening program [27]. The other studies were nationwide (two) [20,21], one multi-center [25], one single-center [28], and the last, a matched case-control investigating the association between combined NCFB and lung cancer in patients with COPD [24].

Most of the studies (*n* = 5) were conducted in Korea [19,23,24,25,28], two were in Taiwan [20,21], and one was in Spain [27].

Participant recruitment involved National Health Insurance Systems (2/8 studies) [20,21], health screening programs (3/8 studies) [19,23,27], and academic medical centers (3/8 studies) [24,25,28]. Follow-up time ranged from 3 years [24,25] to 14 years [21].

All the studies excluded patients who had been diagnosed with cancer before NCFB.

### 3.2. Characteristics of the Population

The total sample size of patients with NCFB was 202,681.

The overall number of NCFB and lung cancer patients was estimated to be 7086. The frequency rates of lung cancer in patients with NCFB ranged from 0.93% (5/354) [27] to 8.0% (4345/53,755) [21]; the mean frequency rate was 2.74% (standard deviation (SD) 3.06; median 1.85%). The highest frequency was observed in Korea and the lowest in Spain.

The mean incidence rate was 3.96 (SD 2.02, median 4.43) [19,20,21,23,24,27]. In two studies, the incidence rate was unknown [25,28].

Patients with NCFB were generally older than 65 years (mean of the population with ≥60 years: 80.5%, SD, 6.67; median 84.10%) [19,20,23], and were mainly males (mean 69.22% of the population, SD, 6, 95; median 67.50) [19,20,21,23,24,25]. Only one study reported the mean age of the population (70 years old, range 61–76) [25], whereas three authors did not report the age of the population [21,24,28].

Three studies [24,27,28], did not report gender. In addition, participants’ ethnic background was never reported.

The measures used by each of the eight studies and related results are detailed in Appendix A.

### 3.3. Association between NCFB and Lung Cancer

Three authors evaluated the risk of developing lung cancer in the NCFB population compared to a non-bronchiectasis cohort.

Choi et al. found an overall higher risk of developing lung cancer in the NCFB population compared to the non-bronchiectasis one (adjusted hazard ratio (aHR) 1.22; 95% confidence interval (CI) 1.14 to 1.30) [19].

Chung, W.S. et al. found that the risk of lung cancer was significantly higher in the NCFB cohort than in the non-bronchiectasis one after adjustment for age, sex, and comorbidities (aHR 2.40; 95% CI 2.22 to 2.60) [19]. The other study from the same team showed similar findings (aHR 2.36; 95% CI 2.19 to 2.55) [21].

Sanchez-Carpintero Abad et al. found a non-significant difference in the incidence and prevalence of cancer in the NCFB population and non-bronchiectasis one. In their analysis, the incidence rates for cancer in groups with and without NCFB were 6.8 and 5.1/1000 person-years, respectively (*p* = 0.62). The differences in proportion for lung cancer diagnosis in baseline study and annual CT scans were 0.84% (95% CI: −0.061–0.23, *p* = 0.256) and −1.84% (95% CI −0.4–0.32, *p* = 0.094), respectively [27].

Some additive factors possibly influencing the incidence of lung cancer were analyzed as well.

### 3.4. Age

Three authors found that older age is associated with a parallel increase in the risk of lung cancer in patients with NCFB.

According to Chung WS, 1-year aging is associated with a 5% of increased risk of lung cancer in patients with NCFB [21].

Similarly, Kim Y et al. showed that the elderly (age ≥ 60 years) had a 3.6-fold risk of developing lung cancer when compared with younger subjects in the NCFB cohort [23].

Chung WS et al. confirmed that the lung cancer incidence rate increased with patients’ age. They found that aHRs of lung cancer in the NCFB cohort compared with the non-bronchiectasis one were significant for all age groups, except for the subgroup aged 20–45 (aHR 5.05; 95% CI 3.36 to 7.60 in 46–55 years, aHR 3.42; 95% CI 2.85 to 4.11 in 46–65 years, and aHR 2.25; 95% CI 1.97 to 2.57 in 65–75 years) [20].

### 3.5. Tobacco Use

Lifetime tobacco use was reported in four out of eight studies [19,23,24,25]. Overall, 49.50% of the patients were either current or former smokers (SD 34.96, median 54.40%). A single study reported the mean smoked pack/year (mean 21.8 pack/year, SD 23.7) [25].

Only one author specified the method used to measure tobacco exposure (self-administered questionnaires) [23].

In the sub-analyses, Choi et al. found that the overall risk of lung cancer was higher in patients with NCFB than those without bronchiectasis regardless of smoking status (aHR for ever smokers 1.26; 95% CI 1.10 to 1.44). However, the risk was significantly higher in those who smoked more than 20 packs/year (aHR 1.14; 95% CI 1.03 to 1.26) but not in those with a history of <10 packs/year (aHR 1.23; 95% CI 0.94 to 1.61) and 10–19 packs/year (aHR 1.21; 95% CI 0.97 to 1.51) [23].

Kim Y. et al. found that current smokers showed an increased risk of lung cancer compared to never-smokers (aHR 3.10, 95% CI 2.00 to 4.79) [23].

Nonetheless, patients with NCFB and a smoking history had the highest risk of all-cause mortality (aHR for lung cancer-related mortality in never-smoker and ever-smoker bronchiectasis patients, respectively, 3.01; 95% CI 1.57 to 5.76, and aHR 14.8; 95% CI 7.66 to 28.61) [28].

### 3.6. Comorbidities

Only a few studies analyzed comorbidities.

Kim Y et al. found that patients with NCFB who developed lung cancer had a significantly higher Charlson Comorbidity Index (CCI) than those who did not develop lung cancer (CCI ≥ 2 in 76.0% in bronchiectasis group vs. 60.5% in controls, *p* < 0.001) [23].

The estimated frequency of COPD in NCFB and lung cancer patients was 41.96% (mean 41.96%, SD 11.46, median 35.50) [19,20,23].

When stratifying the population for the comorbidity COPD, Choi et al. found that the risk of lung cancer was significantly higher in the NCFB group than in the non-bronchiectasis group in patients without COPD (adjusted HR (aHR) 1.19; 95% CI 1.09 to 1.31), but not in those with COPD (aHR 1.36; 95% CI 0.97 to 1.16) [19].

Kim et al. found that the presence of NCFB in COPD patients was inversely associated with the risk of lung cancer (odds ratio, OR 0.25; 95% CI 0.12 to 0.52; *p* < 0.001) [24]. This finding was confirmed in the current/former smoker subgroup (OR 0.27; 95% CI 0.12 to 0.57; *p* < 0.001). Notably, when analyzing the regional association of lung cancer and NCFB in the COPD cohort, one study found that the lobar distribution of NCFB did not differ according to the presence of lung cancer [24]. Another one found that the presence of NCFB was even associated with a significantly lower risk of lung cancer in the same lobe (β value when assuming NCFB as a risk factor of lung cancer: −1.091; 95% CI −1.716 to −0.466; *p* = 0.001) [25].

Chung WS et al. found that the risk of developing lung cancer was higher for patients with NCFB and concomitant COPD (aHR 1.14; 95% CI 1.04 to 1.26) [20].

Asthma prevalence in patients with NCFB and lung cancer was reported in two studies, at 37.0% [19] and 15.7% [20], respectively.

### 3.7. Histological Subtypes of Lung Cancer in NCFB

Histological subtypes of lung cancer in NCFB were reported in three out of eight of the included studies.

Adenocarcinoma was reported with a mean frequency of 36.43% (SD 17.52, median 40.00%), small cell lung cancer with a mean frequency of 16.10% (SD 4.97, median 17.80%), and squamous cell carcinoma with a mean frequency of 15.67% (SD 10.12, median 20.00%) [24,25,27]. In addition, in 7.4% of the cases reported by Kim et al. [25], a poorly differentiated carcinoma was found. In 1/5 (20%) of the cases reported by Sanchez-Carpintero Abad et al., an unknown histological type was identified [27].

Kim et al. found that the presence of NCFB was associated with a lower risk of squamous cell carcinoma in the COPD cohort (OR 0.11; 95% CI 0.03 to 0.49; *p* = 0.001) and no significant association with any other histological type [24]. Furthermore, even after stratifying for smoking status, the authors found that concomitant NCFB was significantly associated with a lower risk of squamous cell carcinoma (OR 0.13; 95% CI 0.03 to 0.61; *p* = 0.009) [24].

### 3.8. Mortality Risk

Sin et al. investigated mortality risk and causes of death in patients with NCFB. They excluded from their analysis all those individuals who had previously received lung surgery and those with other significant diseases which may affect the prognosis (pre-existing cancer, liver cirrhosis, chronic kidney disease, heart failure, acquired immune deficiency syndrome, connective tissue disease, organ transplantation, IPF, pneumoconiosis, lung infections, etc.) [28].

The authors found that the primary cause of death in the NCFB group was malignancy (31.2%), particularly lung cancer (12.4%). In addition, the Cox regression analysis showed that NCFB was significantly associated with increased lung cancer-related death (aHR 3.36; 95% CI 2.18 to 5.18) [28].

Of note, the presence of NCFB was independently associated with a higher all-cause mortality risk than the non-bronchiectasis group (aHR 1.22; 95% CI 1.02 to 1.46), even after adjustment for comorbidities, age, sex, BMI, smoking history, and lung function tests [28].

## 4. Discussion

This systematic review aimed to summarize the evidence on the development of lung cancer in patients with NCFB. Overall, lung cancer occurs in approximately 3% of the patients with NCFB, with higher rates among males and elderly patients. To date, we lack reliable data regarding the epidemiology of lung cancer in NCFB patients. The gender and age distribution of cancer in this population align with the epidemiology of lung cancer in the whole population [33].

Three studies investigated the overall aHR of the occurrence of lung cancer in the NCFB cohort [20,21,27]. However, we could not estimate an overall risk of developing lung cancer due to the heterogeneity of the included populations and cohorts. One small study found no difference in the incidence and prevalence of cancer in the NCFB population and non-bronchiectasis [27]. Interestingly, all the studies assessing the risk of lung cancer in participants with NCFB found a positive correlation even after controlling for smoking history. This adjustment is essential to assume a causal relationship between NCFB and lung cancer independently from a significant confounding factor [34].

Some studies have found that lung cancer risk increases in patients suffering from COPD, asthma, and ILD [35,36,37,38]. Some authors suggested that impairment of mucociliary clearance due to chronic airflow obstruction might increase the exposure of the epithelium to smoking carcinogens, thus facilitating pathologic changes leading to cancer (in particular squamocellular) [39]. Actually, all chronic inflammatory processes causing repeated airway epithelial injury, high cell turnover, and uncontrolled proliferation of cells may lead to tumor formation with an increased risk of cancer [36]. The risk of lung cancer might increase when chronic inflammation is present along with smoking habits.

NCFB shares the same excessive and dysregulated inflammatory activation with asthma and COPD after external stimuli. However, in the case of NCFB, external stimuli are primarily infectious, and the inflammatory pattern is mainly neutrophilic, similar to COPD [40].

Some evidence has also suggested that patients with NCFB are at high risk of developing chronic comorbidities linked to systemic inflammation, including cardiovascular disease, pulmonary hypertension, dyslipidemia, gastro-esophageal reflux disease, osteoporosis, and cancer [3].

Based on these findings, we may assume that chronic inflammation accompanying NCFB explains the heightened risk of lung cancer. Moreover, it is plausible that the risk may be even higher in NCFB patients with a smoking history.

To date, the mechanism of inflammation in NCFB remains poorly defined, and no data regarding the existence of a specific “cancerogenic phenotype” of NCFB is available. Moreover, no scoring system analyzing factors associated with a higher risk of developing lung cancer has been developed for NCFB. This information would be crucial to identify the NCFB at higher risk of lung cancer and to lead the diagnostic and clinical management.

In our analysis, NCFB and COPD co-exist in almost half of the cases, as reported in the current knowledge [41,42]. Despite the known relationship between COPD and lung cancer, the co-existence of COPD on the occurrence of cancer in patients with NCFB remains unclear. Although one might expect a synergistic effect of systemic inflammation due to COPD, smoking, and NCFB, Choi et al. [19] and Kim et al. [24] found that the simultaneous presence of NCFB was associated with a lower risk of lung cancer in the subgroup of COPD patients. The inverse relation between NCFB and lung cancer in COPD patients found by Choi et al. [19], and Kim et al. [24] is quite inexplicable. Similarly, it is not clear why adenocarcinoma is this population is the most frequent cellular (histological) type of lung cancer. We can assume that the strong relation between COPD and smoking habits, and the impact of COPD and smoking on the development of lung cancer [39,43] may have attenuated the effect of NCFB. A relevant factor to consider is the common heavy smoking habit in COPD subpopulations, which may have significantly accentuated the risk of lung cancer (particularly squamocellular) in the group with COPD concerning the group with NCFB alone. This speculation remains challenging to confirm. To point out that the most extensive studies analyzing the association between NCFB and lung cancer do not contain detailed information on smoking history in COPD and non-COPD populations [19,20,21,23,24]. Therefore, we encourage a cautious interpretation of these data, given that the association between NCFB, lung cancer, smoking, and COPD in a large population is not available yet.

Another concern regards the formulation of COPD diagnosis and the selection bias in Choi et al. and Kim et al.’s studies [19,24]. Both authors included COPD patients who underwent a chest computed tomography (CT). Furthermore, they assigned the patients to COPD-cohort utilizing spirometric results. In this context, we may suppose that these patients have more severe diseases and are at higher risk of lung cancer. In addition, mild obstructive patterns are frequently found in many patients with NCFB. We think this may have been a significant confounding factor in the categorization of the patients and the analysis of the impact of NCFB on the development of lung cancer.

Another significant issue to address is the need for more information on the etiology of NCFB. NCFB represents a complex heterogeneous condition, and each underlining cause (COPD, infections, autoimmune diseases, genetic conditions, etc.) is known to impact differently on cancer risk [44,45,46,47]. To date, the chronic inflammatory process in COPD has been extensively analyzed, and different phenotypes associated with specific cytokine patterns favoring the tumor microenvironment have been identified [26]. Conversely, the specific inflammatory pattern has never been investigated in NCFB. Addressing this information gap helps to identify disease-related specific risk factors which may influence the outcome of the patients with NCFB.

Malignancy, specifically lung cancer, is the leading cause of death among NCFB patients. Not surprisingly, Sin et al. [28] also found a significant synergistic effect between NCFB and smoking status on the mortality risk. This finding is consistent with the results from the Bronchiectasis Aetiology Comorbidity Index (BACI), a multi-center cohort analysis which identified malignancies among comorbidities associated with the risk of mortality in patients with NCFB [3].

Finally, we cannot confirm any of the aforementioned hypotheses, given the small sample sizes, the paucity of available studies, and the retrospective nature of the available evidence (Figure 4).

## 5. Strengths and Limits

Our first systematic review focuses on the relationship between NCFB and lung cancer. The systematic nature of the paper significantly limits the risk of missing relevant data due to the exhaustive search strategy. Notwithstanding, this work has some limitations due to the scarcity of available literature and the heterogeneous methodology of the included studies, which did not allow us to perform a meta-analysis. First, all the included studies are retrospective and enrolled patients from single nations. Secondly, nearly half of the participants were drawn from National Health Insurance Systems and health screening programs [19,20,21,23]. Thus, many papers need more essential personal information, such as social status and lifestyle data, smoking habits, occupation, and family history. Third, they suffer a surveillance bias, which may have contributed to the increased frequency of cancer diagnosis in NCFB patients. Notably, three authors reported an augmented aHR of occurrence of lung cancer in the NCFB cohort. However, in all these cohorts, lung cancer diagnosis relied on chest CT according to physician’s suspicion. Therefore, the “no NCFB cohort” might include both actual “no NCFB patients” and those who did not receive the diagnosis of NCFB due to the lack of a CT scan. This bias could significantly overestimate the risk of lung cancer in the NCFB population.

Consequently, we hamper any generalization of these findings. Another significant limit stays in the definitions of NCFB, lung cancer, and concomitant diseases. In half of the studies [19,20,21,23,27], diagnoses were formulated based on health insurance codes, the cancer registry, and the International Statistical Classification of Diseases and Related Health Problems (ICD). Therefore, there might be a significant diagnostic misclassification.

## 6. Conclusions

In conclusion, we found that the presence of NCFB is associated with a higher risk of developing lung cancer than the population without NCFB. This risk is higher for males, the elderly, and smokers, whereas the effect of concomitant COPD is unclear.

The potential association of this heterogeneous condition with lung cancer development represents a serious concern for healthcare systems. However, significant knowledge gaps still exist on this issue. Therefore, well-designed prospective case-control studies and longitudinal multi-national studies are strongly needed to provide more concrete evidence of the association between lung cancer and NCFB. Finally, further analysis might allow for the identification of the subgroups at highest risk.

## Figures and Tables

**Figure 1 life-13-00459-f001:**
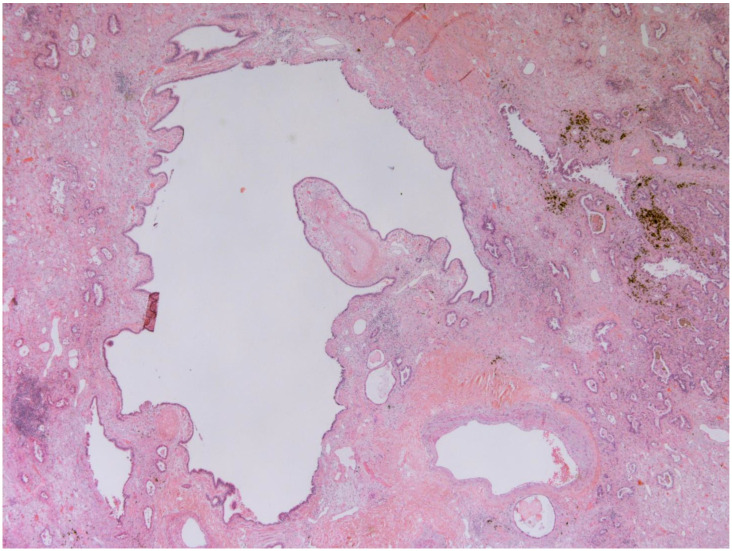
**Bronchiectasis (25x)**: Low magnification shows a hyperectatic, distorted bronchial structure with flattening and focal ulceration of the surface epithelium, fibrosis of the stroma, and inhomogeneous depletion of the muscular and cartilaginous components. The pertinent artery shows thickening of the media suggestive of possible small circle pressure disorder. The bronchus is surrounded by cytoarchitecturally atypical glandulomorphic structures consistent with adenocarcinoma (acinar and lepidic pattern). The images have been provided free of charge by the Laboratory Department of Medical and Biological Science, Azienda Sanitaria Universitaria Integrata di Udine, Italy, and have been taken by Dr. Alessandro De Pellegrin.

**Figure 2 life-13-00459-f002:**
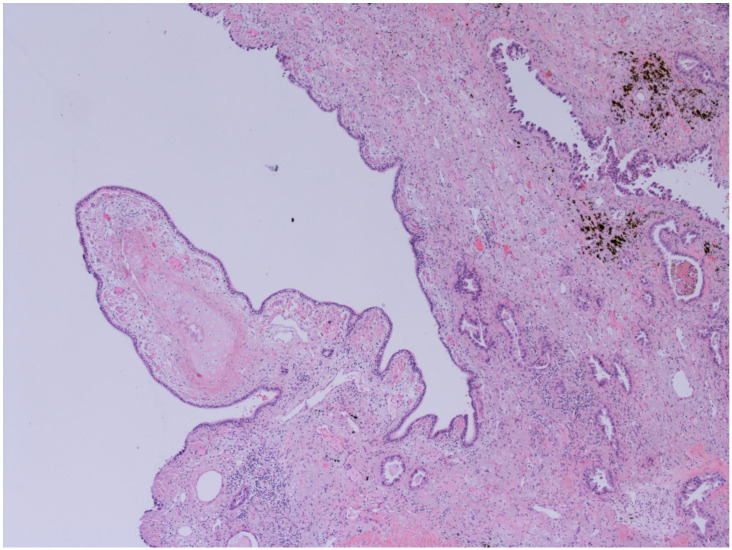
**Bronchiectasis and Lung adenocarcinoma (100x):** Higher magnification highlights carcinomatous gland infiltration of the bronchial stroma, which shows chronic inflammatory infiltrates, dense fibrosis, and microcystic atrophy of the peribronchial glands; the surface epithelium is flattened and eroded. The images have been provided free of charge by the Laboratory Department of Medical and Biological Science, Azienda Sanitaria Universitaria Integrata di Udine, Italy, and have been taken by Dr. Alessandro De Pellegrin.

**Figure 3 life-13-00459-f003:**
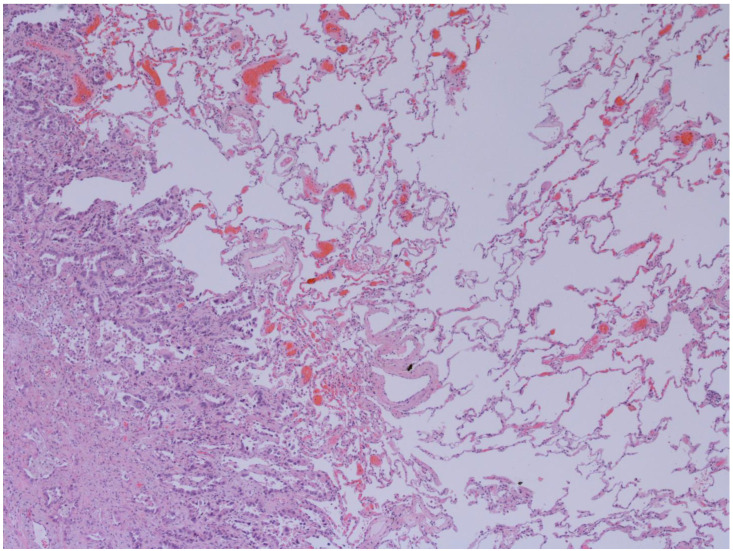
**Emphysema and Lung adenocarcinoma (50x):** Interface between lepidic pattern adenocarcinoma in sclero-elastotic stroma and parenchyma with emphysematous changes. The images have been provided free of charge by the Laboratory Department of Medical and Biological Science, Azienda Sanitaria Universitaria Integrata di Udine, Italy, and have been taken by Dr. Alessandro De Pellegrin.

**Figure 4 life-13-00459-f004:**
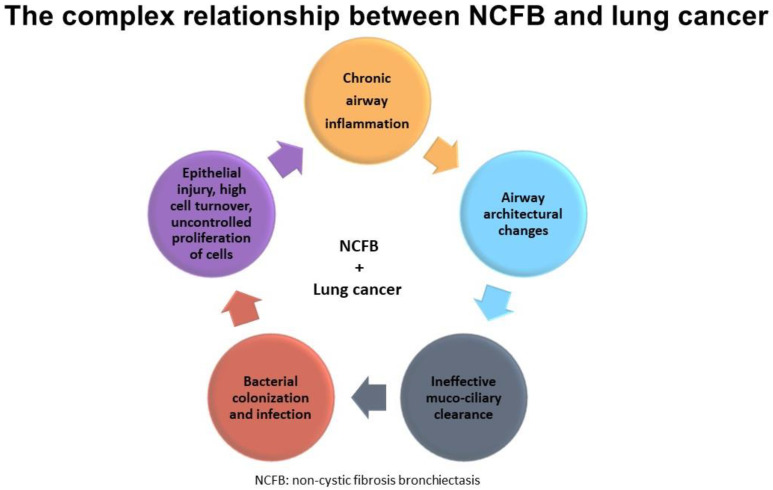
Shows the complex relationship between NCFB and lung cancer.

## Data Availability

Not applicable.

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
