# Peer review of "Do Patients with Bronchiectasis Have an Increased Risk of Developing Lung Cancer? A Systematic Review"

_life, 2023, doi:10.3390/life13020459_

Round 1

Reviewer 1 Report

The chronic respiratory disease patients have been always at a risk of developing lung cancer. Adult non-cystic fibrosis bronchiectasis (NCFB) is the third most common chronic inflammatory respiratory disease, after chronic obstructive pulmonary disease (COPD) and asthma. The authors approach to conclude that NCFB patients are more prone to develop lung cancer especially elderly males with smoking habits; is appreciable. 

Major Comments:

1.     It would have been great, if you could provide the representative histology images of NCFB, COPD and adenocarcinoma.

2.     Please also include a paragraph discussing critical molecular signaling pathways involved in bronchiectasis and probable therapeutic targets. 

Minor Comments:

1.     Please summarize the entire Results section in a comprehensive table format for easy understanding.

2.     Provide a graphical abstract or figure explaining this complete systematic review. 

Reviewer 2 Report

The authors conducted a systemic review to evaluate the risk of lung cancer in bronchiectasis patients. And concluded the overall risk of developing lung cancer in the bronchiectasis population was two folds than the non-bronchiectasis one. Males, the elderly, and smokers had a higher risk of developing lung cancer. However, several major points need to be addressed

1.     The enrolled 4 papers about bronchiectasis and lung cancer were all retrospective and come from different programs, In Taiwan, Chung et al enrolled 53,755 patients newly hospitalized with bronchiectasis between 1998 and 2010 using data from the Taiwan National Health Insurance Research Database, the comparison cohort comprised 215,020 people from the general population without bronchiectasis, the same cohort studies in the same time but had  different results, aHR 2.40( 95% CI 2.22 to 2.60) and (aHR 2.36; 95% CI 2.19 to 2.55) ?. In Choi et al, they used the population-based cohort study of 3,858,422 individuals who participated in the 2009 National Health Screening Program in Korea. Maria Sanchez-Carpintero Abad et al enrolled 3028 individuals participating in an international multicenter lung cancer screening consortium (I-ELCAP) were selected from 2000 to 2012 and they used the incidence rates for cancer in groups with and without NCFB were 6.8 and 184 5.1/1000 person-years, respectively (p = 0.62). The 3 studies enrolled different populations and cohorts, the authors cannot put them all together and make a conclusion that an estimate an overall comparative risk of developing lung cancer in NCFB. These huge heterogenous made the result is not reliable.

2.     Authors cited 4 papers published by three authors to conclude lung cancer significantly developed in NCFB than in non-NCFB patients. The diagnosis of lung cancer always relied on chest CT, and “no NCFB” may enroll those who are true “no NCFB” and who are NCFB but they didn’t receive chest CT. For example, In Chung et al in Taiwan, we only know the association between NCFB and lung cancer, we cannot make sure of the diagnosis of NCFB from those who received chest CTs based on lung cancers suspected by physicians. The phenomenon made the risk of lung cancer overestimated in the NCFB population. Authors had to enroll more well-defined and homogenous studies and make a reasonable result.

3. Words spelling (in tables) needs to be reviewed by a native English

Reviewer 3 Report

1. Some sentences have multiple references for example in page 2, line 45 and line 50 and ....

Please reform this multiple references and put proper references at the end of each sentence

2.Some sentences of the manuscript doesn't have appropriate reference. For example in page 2, line 46-48, the sentence " The clinical overlap ... and underdiagnoses of NCFB"

Please add proper reference at the end of each sentence of the manuscript (exept sentences that are well-established or the results of present panuscript)

3.In page 1, line 114; what does the abbreviation "NCFB" stands for?; please define it

4.In page 1, line 15-16 and line 18-19, authors mentioned "standard deviation" and "median".

Please note that the writers should not mention detailed statistics like "standard deviation" and "median". Thus, reconsider these two word.

5.In page 1, part" Simple Summary"; the authors should keep continuity and simplicity of this part. For example, in line 10-11, the authors have written about "the important accompany of chronic lung conditions with lung cancer" but in the next sentence, the continuity is disrupted and writers have talked about their current work.

Please reconsider this part according to mentioned note.

On the other side, the authors have remarked their results a little complex; please reconsider the part"Simple Summary" and transform it into a simple part so that other readers can understand it easily.

6.In page 1, part" Simple Summary", the authors should mention a little about the conclusion of their current study. Please exert this note in your manuscript.

7.In page 1, line 23-25, the sentence" We conducted a systematic review to ... who develop bronchiectasis-associated lung cancer. " is repeated (this sentence is mentioned in the part" Simple Summary"

Please reform mentioned sentence in order to avoid self-plagiarism

8. In page 3 and 4, line 133-154; please create a separate part called" Characteristics of included studies" in the part" Results" and transfer mentioned part of the manuscript to it.

9.In page 6 and 7, line 291-294; the paragraph" It is widely accepted ... including lung cancers [31]." disrupts the continuity of 

the text. Please tranfere it to the proper place of the manuscript or reconsider it.

10.In page 6, line 278-282, the paragraph" This systematic review aimed to ... males and elderly patients." doesn't have proper continuiry. Moreover, mentioned paragraph has some results without enough discussion. Please reconsider or omit it.

11. In page 7- 8, line 346-349, the sentence" Finally, it is impossible to confirm ... of the available evidence." belongs to the end of the part "Discussion". Please move it to mentioned part.

12. In the part" 6. Conclusions" 

authors should write about the conclusion of their work which support the maintitle of manuscript. Extra information about findings should be written in the part "Discussion". Thus, please reconsider line 376-387 (omit or rewrite it)

Round 2

Reviewer 2 Report

Thanks for the authors’ kind reply. Based on the retrospective entity of the studies, selection bias is inevitable. I agreed with the authors who wrote additional warning words in the limit section to avoid overestimating the risk of lung cancer in bronchiectasis patients.